# Innovative Development of Pasta with the Addition of Fish By-Products from Two Species

**DOI:** 10.3390/foods10081889

**Published:** 2021-08-15

**Authors:** Andrea Ainsa, Adrián Honrado, Pedro L. Marquina, Pedro Roncalés, José Antonio Beltrán, Juan B. Calanche M.

**Affiliations:** Agroalimentario de Aragón-IA2, Universidad de Zaragoza-CITA, Miguel Servet, 177, 50013 Zaragoza, Spain; andreaainsa6@gmail.com (A.A.); adrihonfri@gmail.com (A.H.); pmarquin@gmail.com (P.L.M.); roncales@unizar.es (P.R.); jbeltran@unizar.es (J.A.B.)

**Keywords:** enriched pasta, bioactive compound, tuna, sea bass, Ω-3 fatty acids, sensometrics, TPA (texture profile analysis)

## Abstract

The fish industry generates by-products that are still nutrient-rich. Its incorporation in pasta production could be an interesting option to get functional food. Therefore, the aim of this study was to compare the nutritional composition, technological properties and sensory quality of two pastas containing tuna and sea bass by-products, separately. Durum wheat semolina and fish by-product concentrates were used in pasta manufacturing. Fatty acids profile, optimal cooking time, texture profile analysis, color, weight gain, swelling index, cooking losses and moisture were determined and compared with a non-containing fish reference. A sensory analysis was also carried out. In general, results showed a higher content of fatty acids in tuna pasta than in sea bass pasta. The texture profile analysis (TPA) showed lower hardness and fracturability in the fish pasta. Cohesiveness was higher in the tuna pasta while sea bass pasta was brighter. Fish incorporation caused a decrease in weight gain and swelling index and an increase in cooking losses. Sensory analysis established differences in homogeneity, typical aroma, fish flavor, fish odor and elasticity. It was concluded that the use of these by-products results in a more nutritious pasta although tuna content should be reduced (<3%) to improve its sensory profile.

## 1. Introduction

An essential component of modern society is the consumption of functional foodstuffs and nutraceuticals [1]. Fundamentally, the main purpose among consumers to intake these foods is the aim to reduce the risk of chronic diseases or to enhance health [2]. The biological effect obtained from functional foodstuffs is related to different compounds which are naturally present or intentionally added to the product. These products can be classified as fortified, increasing the content of a natural component, or enriched, adding an external component. In this sense, important nutritional components are incorporated in different foods, such as Ω-3 fatty acids, amino acids (e.g., arginine, leucine, and tyrosine), and minerals (calcium, phosphorus or manganese among others) [3].

Pasta seems to be an excellent opportunity to be enriched by incorporating alternative bioactive compounds as it is a common food due to its easy handling, storage, and preparation, in addition to its low cost [4,5]. Considering that fish contains great values of Ω-3 fatty acids, pasta enriched with fish would be a good chance to achieve the suggested daily intake of healthier fatty acids and it could be defined as high content of Ω-3 fatty acids according to Council Regulation (CE) N° 116/2010 [6,7,8]. Fish has a high content of protein, essential amino acids, and it is a great source of vitamins (A, D, B6 and B12) and a wide variety of minerals (phosphorus, magnesium, iron, zinc, and iodine) [9,10].

The energy value of fish is largely determined by their content of lipids. This fat is a relevant source of bioactive compounds. The evaluation of farmed sea bass by-products indicates that it is feasible for the use of this fish to obtain MUFA, PUFA, Ω-3 fatty acids, minerals, proteins and amino acids [11]. However, blue fish contains more Ω-3 fatty acids, DHA and EPA than white fish [12]. For this reason, tuna (*Thunnus obesus*) from fishing is an excellent source of high-quality protein and Ω-3 polyunsaturated fatty acids [13].

Several researchers have evaluated the importance and potential use of food industry by-products [14,15], which could also reduce the environmental impact of this industrial sector. The development of strategies to take advantage of food industry by-products is a necessary action to improve the efficiency of industrial operations, reduce waste, and recover high-added value compounds for further utilization in food products which is named as circular economy [16]. The action plan of circular economy established measures covering the whole life cycle: from production and consumption to waste management and the market for secondary raw materials and a revised legislative proposal on waste [17]. It is intended to achieve some of the sustainable development goals among which are: (2) zero hunger, (3) good health and well-being, (12) responsible consumption and production or (14) life below water [18].

The present study was carried out to compare the effect of two different fish by-products from a distinct origin (fished and farmed) in pasta-making, assessing their nutritional values, technological properties, and carrying out a sensory study. The main purpose was to achieve a great enrichment in ALA, EPA and DHA through the addition of fish.

## 2. Materials and Methods

### 2.1. Raw Material

Sea bass (*Dicentrarchus labrax*) by-products (flawed fillets and flesh cut) from aquaculture were supplied by a local fish industry (Scanfisk^®^, Zaragoza, Spain) and tuna (*Thunnus obesus*) by-products (head and flesh cut) were supplied by the local fishermen of Pontevedra, Spain. With respect to the cereal source, semolina from durum wheat (*Triticum durum*) was provided by a local company (Pastas Romero^®^, Zaragoza, Spain). The antioxidant used was rosemary extract powder (E-392) provided by Marbys^®^ (Barcelona, Spain).

### 2.2. Enriched Pasta Preparation

The concentrates (≈18.5% moisture) were produced in the same way. Frozen fillets and cuts from sea bass and tuna (by-products) were manually deboned meat (MDM) and processed according to the methodology described in our previous studies [19,20]. Two types of pasta with durum wheat semolina (≈11.7% moisture) were made from each species used. They were produced with an experimental extrusion machine (Imperia & Monferrina, Mod. P6 LM14040, Moncalieri, Italy) in *fusilli* format according to Calanche et al., 2019. Enriched pasta with both fish concentrates was desiccated to obtain dry pasta (≈10% moisture) that was stored at room temperature. A control pasta composed of durum wheat and water was manufactured as a comparison for some of the analyses. Pasta formulations and their proximal analyses are shown in Table 1.

### 2.3. Fatty Acids Profile

Fatty acids profile was determined according to Bligh and Dyer method (1959) [19], taking into account modifications made in the analysis protocol in agreement with the procedure described by Ainsa et al. (2021) [20] to get a better adjustment to assayed enriched pasta. Each sample was homogenized with different solvents (chloroform, methanol, potassium chloride and water) using an Ultraturrax device (IKA-WERKE, T-25 basic). Subsequently, it was centrifuged at 4000 rpm at 10 min, and the fat was extracted. Solvents were evaporated with BHT (butylated hydroxytoluene) as an antioxidant. Then, 2 mL of hexane and 1 mL of potassium hydroxide saturated were incorporated. Fatty acid profile was analyzed using a gas chromatograph (HP-6890II). Fatty acids were measured as the total area (%) of identified fatty acids.

### 2.4. Optimal Cooking Time

The optimal cooking time was estimated with a Warner–Bratzler cut test according to the instruction manual of the texturometer used (ANAME, TA-XT2i). The determination of the optimal cooking time was made using the texturometer with a flat Warner–Bratzler device. The instrumental measure of hardness was carried out in cooked pasta according to sample times assayed. Hardness was defined as the maximum force (tangential angle) required to cut the sample and was expressed in kg/mm*s^2^. Test conditions development was: pre-test speed: 2 mm/s; test speed: 2 mm/s; post-test speed: 10 mm/s; cutting distance: 15 mm; threshold strength: 0.010 kg. The hardness of pasta was determined by triplicate.

### 2.5. Texture Profile Analysis—TPA

A texturometer (ANAME, TA-XT2i) with a cylindrical flat aluminum probe was used for texture profile analysis (TPA). The method consisted of the application of two compression cycles with decompression of 20 s over cooked pasta. In this way, it was possible to determine hardness, adhesiveness, springiness, cohesiveness, gumminess, chewiness and fracturability. The conditions were: test speed: 2 mm/s; sample deformation: 75%; force threshold: 10 g. Five measurements were made for each type of pasta.

### 2.6. Pasta Color

Color analysis in cooking pasta according to its optimal cooking time was made using a colorimeter (Minolta. CM-2002, Japan). The CIE L*a*b* system represented by L* (brightness), a* (redness) and b* (yellowness) was used. The color variation produced by each fish species was calculated with a total color difference (∆E) between control pasta and sea bass and tuna pasta:(1)$$\Delta E=\sqrt{{\left(\Delta L^{*}\right)}^{2}+{\left(\Delta a^{*}\right)}^{2}+{\left(\Delta b^{*}\right)}^{2}}$$
ΔL* = L* Fish pasta- L* Control pasta; Δa* = a* Fish pasta- a* Control pasta and
Δb* = b* Fish pasta- b* Control pasta

### 2.7. Technological Properties

#### 2.7.1. Weight Gain and Swelling Index

The weight gain was established with 3 g of pasta which was cooked in 180 mL of water during optimal cooking time, they were cooled in 100 mL of water; then, pasta was dried superficially with absorbent paper and weighed on an analytical balance [21]. This parameter was calculated from the following formula:(2)$$\mathrm{WG}\left(\%\right)=\frac{Weight\;of\;cooked\;pasta-Weight\;of\;cooked\;pasta\;after\;drying}{Weight\;of\;cooked\;pasta\;after\;drying}\cdot 100\;$$

Then, the cooked pasta was dehydrated in an oven at 105 °C for 24 h. The swelling index was determined by the following equation:(3)$$SI=\frac{Weight\;of\;cooked\;pasta\;\left(g\right)}{Weight\;of\;dried\;cooked\;pasta\;\left(g\right)}\;$$

#### 2.7.2. Cooking Losses

A sample of 3 g of each pasta was added in 180 mL of water and cooking during the optimal cooking time [22]. The water resulting after cooking was collected in crucibles and evaporated on a stove at 105 °C until reaching a constant weight. The residue was weighed and determined as a percentage of the total weight of raw pasta.

#### 2.7.3. Moisture

Moisture was evaluated by a gravimetric method. The pasta was weighed and then dried in an oven at 105 °C until reaching a constant weight. It was cooled to room temperature and weighed again.
(4)$$Moisture\;\left(\%\right)=\frac{raw\;pasta\;weight-dried\;pasta\;weight}{raw\;pasta\;weight}\cdot 100$$

### 2.8. Sensory Analysis

A panel of ten selected assessors with previous experience in sensory analysis of fish and pasta bellowing to the staff of Meat Science and Technology Official Research Group (A04_20R DGA) from the University of Zaragoza was used to carry a sensory method knowing as “deviation respect to a reference” -DR- [23] which uses a reference sample (control food) against all evaluated samples (assessed food). The assessors had demonstrated sensory sensitivity in preliminary tests, received considerable training and they were able to make consistent and repeatable sensory assessments of various samples of pasta. The panel received prior training with respect to the use of the DR method and intensity scales to evaluate different attributes in pasta according to requirements of ISO standards (ISO 8586: 2012) [24]. Along this process, panelists became familiarized with the different descriptors and their intensity scales in order to assess the samples in a more accurate form [25]. The attributes selected for this study were: homogeneity, characteristic color, typical aroma, fish odor, rancidity flavor, hardness, elasticity, pastiness, pasta characteristic flavor, fish flavor and after-taste. All of them are based on previous studies [19,20].

Once the panel was prepared, the trained assessors indicated the degree of difference in intensity for each sensory attribute using a non-structured lineal scale of 10 cm anchored to the extremes as “none” to “much”. The pasta was prepared by boiling until the optimal cooking time previously established and was served without any accompaniment at 60 °C according to Standard UNE-ISO 6658:2019 [23]. Each enriched pasta was served in an independent trial together with the reference pasta (control) and both were evaluated at the same time.

### 2.9. Statistical Analyses

Results of this study were analyzed using an XLSTAT Version 2016 (Addinsoft^©^, Paris, France). A univariate analysis was performed to check the normality of the data and detect outliers. Then, statistical analysis was performed by simple ANOVA (types of pasta) and Fisher test with a 95% confidence interval was used a posteriori to find differences among means for physical and chemical measures. To get a comparison between TPA and fatty acids content, a Pearson correlation was made and then, principal component analysis (PCA) was performed to explore relationships or associations that were of interest for this set of variables. In the sensory analysis, panel analyses were performed to establish the reliability of the results, verifying the panel’s performance as well as its discriminative power. Posteriorly, ANOVA was performed to obtain significant differences with respect to control using a Dunnett test *a posteriori* (95% confidence interval) to establish differences between each type of pasta and control pasta. Then, the Fisher test was performed to find differences among all pasta (control/tuna/sea bass). Finally, characterization of each pasta was made with the square cosine method to get sensorial profiles, which were represented in a biplot where confidence ellipses of Hoteling (95%) were drawn to compare the samples.

## 3. Results

### 3.1. Comparison of Fatty Acids Profiles

The fatty acid profiles for both types of pasta is shown in Table 2.

Saturated fatty acids percentage was significantly higher in tuna pasta than in seabass pasta. However, monounsaturated fatty acids were higher for sea bass pasta, especially due to oleic acid. Concerning polyunsaturated fatty acids, there were no significant differences between both percentages although EPA and DHA contents were higher in tuna pasta. The values of Ω3 and Ω6 presented a contrary behavior, while Ω3 value was higher in tuna pasta, Ω6 was higher in sea bass due to the content of linoleic acid.

Regarding fatty acids ratios, P/S and Ω6/Ω3 ratios were significantly higher in pasta enriched with sea bass. Nevertheless, the Ω3 content (mg/100 g) in pasta with tuna was almost double that in pasta with sea bass, thus, it presented a significant difference. The percentage of DRI (dietary reference intake of Ω3) was higher in tuna pasta because of the Ω3 content.

### 3.2. Optimal Cooking Time

The behavior about optimal cooking time for tuna and sea bass pasta was similar. Hardness reached an inflection point which is considered to be the optimal cooking time. Therefore, the perfect time for cooking was 210 s for pasta with fish while for control pasta was 390 s due to the significant decrease in hardness as shown in Figure 1. Although the time was the same for both kinds of pasta with fish added, tuna pasta was harder than sea bass pasta.

### 3.3. Texture Profile Analysis -TPA-

As can be seen in Table 3, control pasta presented significantly higher hardness (*p* < 0.05) than enriched pasta with fish being significantly lower (*p* < 0.05) in tuna pasta. Related to fracturability, being higher in sea bass than tuna pasta. However, cohesiveness had a significantly higher value (*p* < 0.05) for tuna pasta which was similar to control pasta. On the other hand, gumminess and chewiness had a similar behavior being higher in control pasta than in pasta with fish. Related to adhesiveness, sea bass pasta seemed to have the same behavior as control pasta.

For a better understanding of the relationship between physical and chemical variables, a PCA using a Pearson correlation matrix was developed. In essence, a multivariate analysis is a tool to simultaneously find patterns and relationships among several variables. It allows us to predict the effect that a change in one variable will have on the other variables [26]. In this regard, between TPA values and the fatty acid composition for each type of enriched pasta, a relationship could be detected for some parameters as could be seen in Figure 2.

Related to sea bass (Figure 2A), a correlation was shown for cohesiveness, gumminess and chewiness (F1). Cohesiveness showed a negative correlation with saturated, *trans* and polyunsaturated fatty acids while had a positive correlation (>0.89) with monounsaturated, especially oleic and linoleic showing up close at the PCA. Gumminess had a similar performance having a negative correlation with saturated and trans fatty acids. However, a positive correlation was found with polyunsaturated fatty acids showing on the right side of the PCA. Finally, chewiness had the same behavior as gumminess.

Regarding tuna pasta (Figure 2B), only chewiness had a significant correlation with some fatty acids. It was related to monounsaturated and polyunsaturated fatty acids whereas its behavior was contrary to saturated fatty acids. However, this parameter showed a negative correlation with polyunsaturated fatty acids (DPA -C22:5 Ω-3- and EPA) showing on the right side of the PCA.

### 3.4. Pasta Color

The color parameters are shown in Figure 3.

Related to luminosity (L*), control and sea bass pasta presented similar values which are higher than in tuna pasta. The opposite effect was observed in the red index (a*) being significantly higher (*p* < 0.05) in tuna pasta than in control and sea bass pasta, which shows similar values. In the case of the yellow index (b*), pasta with fish concentrate had the same behavior between them and control pasta had higher values than enriched fish pasta. Finally, the total color difference (∆E) showed that it in tuna pasta was higher than sea bass pasta (*p* < 0.05), confirming a large effect over the color which depends on the species used.

### 3.5. Technological Properties

Values of technological parameters are shown in Table 4. The addition of fish in pasta formulation showed a significant decrease in weight gain (WG) with similar behavior in sea bass and tuna pasta. Related to the swelling index (SI), there were significant differences among all pasta assayed, the control pasta had the highest index and tuna pasta showed the lowest value. In the same way, sea bass and tuna pasta did not show differences in cooking losses (CL) while the control was significantly distinct from the rest. Finally, moisture (M) was higher (*p* < 0.05) in the control pasta while both kinds of enriched pasta had a similar value.

### 3.6. Sensory Analysis

A radial graph (Figure 4) was made to compare the intensity of the studied attributes for each kind of enriched pasta compared with durum pasta (control).

According to ANOVA, significant differences were found for sensory attributes such as homogeneity, typical aroma, elasticity, fish flavor and fish odor. The control pasta had the highest value to homogeneity (9/10) while enriched pasta had a similar score (≈7/10). Typical aroma showed the same trend with a higher value for control pasta (8/10) than both enriched pasta. As in previous parameters, control pasta showed a higher value for elasticity (below 7). Regarding fish odor, as expected, the control pasta did not show this attribute whereas tuna had the highest value (6/10) followed by sea bass pasta (4/10). Related to the above, fish flavor was not present in the control pasta while sea bass had a lower value (≈1/10) than tuna pasta (5/10).

Sensory profiles of the pasta assessed are shown in Figure 5. Based on discriminatory power for each attribute assessed by the sensory panel only 8 of 11 were selected to draw profiles that turned out to be very different from each other. The first component (F1) collected 82.20% of the total variation and shows a clear separation between control pasta and enriched both kinds of pasta. Tuna was characterized by its fish odor (*p* < 0.01), fish flavor (*p* < 0.01) and after-taste. In contrast, the control pasta showed a typical aroma (*p* < 0.05) and it was found close to pasta characteristic flavor and elasticity (*p* < 0.01). However, in enriched pasta with sea bass no particular attribute stood out, being located in the plot between the control pasta and the tuna pasta. Due to their proximity, enriched pasta could resemble each other.

In order to get a relation between sensory analysis and fatty acids composition in enriched pasta, a PCA was made and is shown in Figure 6.

The first component (F1) collected 48.08% of the variability and separated the different types of fish used. Tuna was related to fish odor and fish flavor and, with some polyunsaturated acid, in special those belonging C18 type, as well as C20:2 n-6 and C22:5 n-3. Additionally, rancidity odor, hardness and after-taste were associated with tuna too. The after-taste was highlighted due to its relationship with the *trans* C18:2 n-6. Fatty acids from cereals, (C18:2 n-6, C18:1 n-9 and C18:3 n-3) located on the left side of the plot and close to Tuna were related to rancidity odor. Conversely, on the other side of the plot, the sea bass was related to a typical aroma and characteristic flavor of the pasta. For its part, homogeneity, elasticity, and pastiness showed similar behavior and were located in the middle of the plot. The most outstanding attributes of the seabass pasta were associated with saturated fatty acids (C14, C15, C16, C17, C17, C20 and C22) and some polyunsaturated fatty acids as EPA (C22:6 n-3) and DHA (C20:5 n-3) which could be associated with the characteristic flavor of this kind of pasta.

## 4. Discussion

### 4.1. Fatty Acids Profile

As expected, pasta with tuna concentrate had a higher content of fatty acids than in sea bass concentrate, although it had some exceptions. Due to the last changes in the aquaculture feed, the increase in vegetable oils in the diet of Mediterranean farmed fish caused the oleic and linoleic content to be higher in pasta with sea bass than tuna, which resulted in an increase in %MUFA and content of Ω6 [24]. As seen in other studies, the content of EPA and DHA, the most important for a healthy diet [27], was higher in tuna, especially for the species used in this study (*Thunnus obesus*) [28,29]. Therefore, the behavior observed in these fish could be seen in pasta enriched with them. Concerning ratios, although the ideal ratio for Ω6/Ω3 was considered to be around 4:1 [30], this study found values higher in sea bass pasta. Some previous studies found ratios that agreed with our results [5,19]. On the other hand, EFSA made a recommendation of 300 mg for the Ω3 mg/100 g ratio for a day [31]. With the tuna pasta developed, we reached (80.97%) of dietary reference intake of Ω3, while with the sea bass pasta, it was reached 44.75% mg/100 g). Consequently, our findings demonstrated that enriched pasta with tuna or sea bass represented an adequate source to get the daily reference intake of PUFA (%DRI) reaching almost 81% for tuna pasta and 45% for sea bass pasta [32].

### 4.2. Cooking Times

As could be seen in the results, enriched pasta needed a lower time for cooking than control pasta. When fish is incorporated into pasta formulation, Physico-chemical characteristics are modified. The starch content decreases and for this reason, the water required for its gelatinization decreases too. The substitution of semolina implies a decrease in the glutenin content. Thus, less time to cook is required [33]. On top of that, the required time was higher for enriched pasta than in a previous study due to the percentage of fish in the formula [20].

### 4.3. Texture Profile Analysis

The effect of different added components such as starch, lipid or other ingredients could influence the pasta texture profile, especially for hardness [32]. In this way, the decrease in hardness parameter detected in enriched pasta was associated with the weakening of the structure due to the incorporation of lipids and proteins from fish meat that modify the matrix of gluten and starch [33]. Regarding fracturability, it is usually associated with hardness and was different between enriched pasta and control. It could be due to the behavior provided by myofibrillar proteins but especially by fat composition in each kind of pasta. According to other studies, texture properties could be modified with the incorporation of other ingredients different from semolina and resulted in unwanted textures [34]. Concerning cohesiveness, tuna and control pasta had the same values, whereas, in adhesiveness, sea bass and control were similar. These findings confirm the behavior of *trans* fatty acids like saturated fatty acids instead of other unsaturated in sea bass pasta. The above is a common characteristic of vegetables processed oils widely used in the animal feed industry. Conversely, tuna, a fish from the catch, had a higher value of adhesiveness which may be due to its unsaturated fatty acids quantities.

### 4.4. Color of Pasta

Brightness and red index (a*) had the same behavior. The incorporation of sea bass in pasta did not modify these parameters in comparison with enriched pasta with tuna due to the difference of color between these fish species. However, the yellow index (b*) was one of the most important parameters for pasta acceptability [35] and it was a characteristic color of pasta. For this reason, control pasta had the highest value for this index. The global color variation (∆E) with respect to control pasta was higher in tuna pasta. Sea bass showed values in agreement with another study [9]. The results had variations in the range 3–6 as can be seen in Figure 3. Despite this, these changes cannot probably be seen with the naked eye [9].

### 4.5. Technological Parameters

The capacity of pasta to absorb water depends on its composition and processing conditions [36]. In this way, lipids and proteins from fish interact with starch for water absorption and reduce starch hydration, thus, enriched pasta had lower weight gain. On top of that, the reduced swelling index could be due to the formation of a protein network and different complexes between starch and lipids. These results were found in other studies [9,37]. On the other hand, the increase in cooking loss could be produced by the introduction of non-gluten proteins that weakened the network structure. Similar behavior had been shown in other studies of pasta fortified with other ingredients [4,9,38]. Finally, the values obtained for moisture were below those marked by regulation, which sets 12.5% of moisture for dry pasta [39].

### 4.6. Sensory Analysis

The addition of fish concentrates to enrich pasta caused some changes in their sensory profiles, especially providing odor and taste, as noted in previous studies [19,20]. This fact explained the increase in fish odor and flavor in enriched pasta being higher in tuna which presented more unsaturated fatty acids. Organoleptic properties obtained in this study are in agreement with those results reported by Devi (2013) [40] in pasta with incorporated fish. Furthermore, there were no remarkable changes in texture and appearance attributes due to the low percentage of fish concentrate added (3%). Regarding the above, earlier research that used tuna and tilapia mill meat to making laminated pasta (lasagnas) demonstrated there were no significant differences in these aspects either [41]. According to Figure 3 and Figure 4, sea bass had an intermediate profile between control pasta and tuna pasta. It was similar to control in yellow color, pasta characteristic aroma and elasticity but differs from tuna pasta in the quantities of unsaturated fatty acids, especially Ω3 (EPA and DHA), being higher in tuna and therefore offering a typical smell and taste of fish. The above was corroborated in Figure 6 where fishy odor and flavor were related to fatty acids of tuna pasta.

In summary, it is possible to affirm that pasta could be enriched with both species of fish (tuna and seabass). However, the pasta profile depends on the type of fish used because each one provides completely organoleptic properties due to its composition in fatty acids as a consequence of its origin and diet.

## 5. Conclusions

The use of sea bass and tuna by-products to enrich pasta is an excellent alternative due to the contribution of protein and polyunsaturated fatty acids (Ω3) in these species. Tuna contributes three times as much DHA and EPA to pasta as sea bass. Therefore, the addition of tuna could be reduced from 3% to 1% in future studies to improve its sensory profile, which is characterized by a high fish flavor and fish odor compared to sea bass pasta and control pasta. Besides this, texture profile, color and technological quality parameters were modified by the addition of fish. The texture parameters showed a significant decrease in almost all parameters compared to control pasta as in the technological parameters, except for cooking losses, which were higher for enriched pasta than for control pasta. Finally, the sensory profiles of all pasta were adequate, showing a better behavior in sea bass pasta in comparison with control pasta.

## Figures and Tables

**Figure 1 foods-10-01889-f001:**
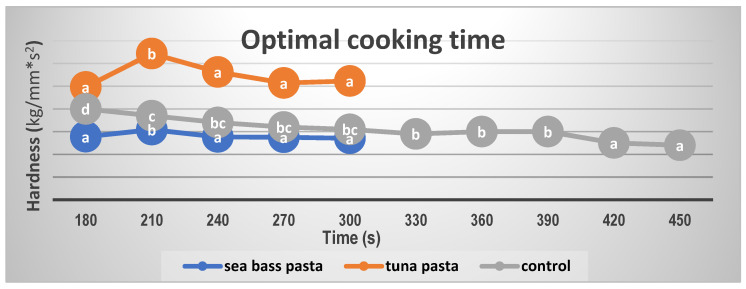
Optimal cooking time for tuna and sea bass pasta. Distinct letters indicate significant differences (*p* < 0.05) among the cooking times for each type of pasta.

**Figure 2 foods-10-01889-f002:**
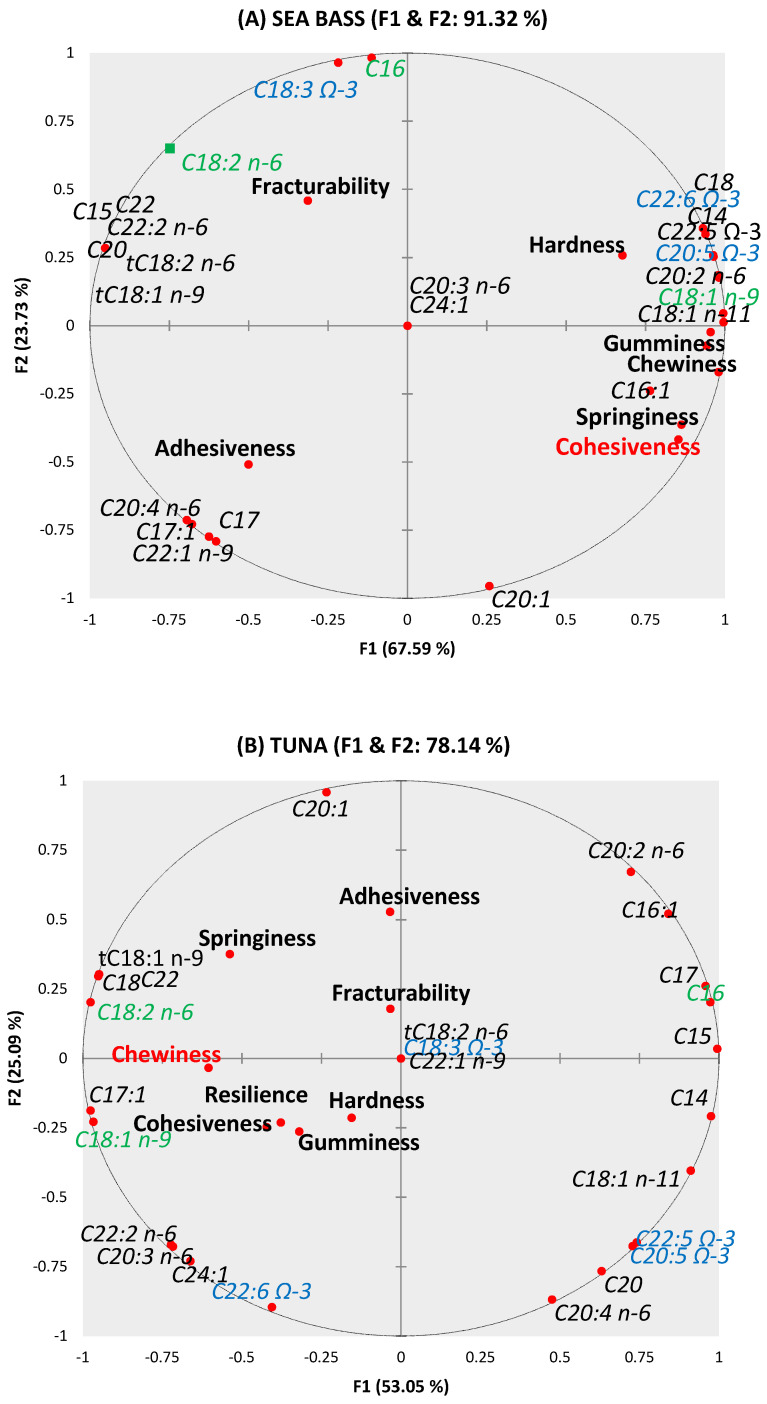
PCA of correlations between TPA parameters and fatty acids composition for sea bass pasta (**A**) and tuna pasta (**B**).

**Figure 3 foods-10-01889-f003:**
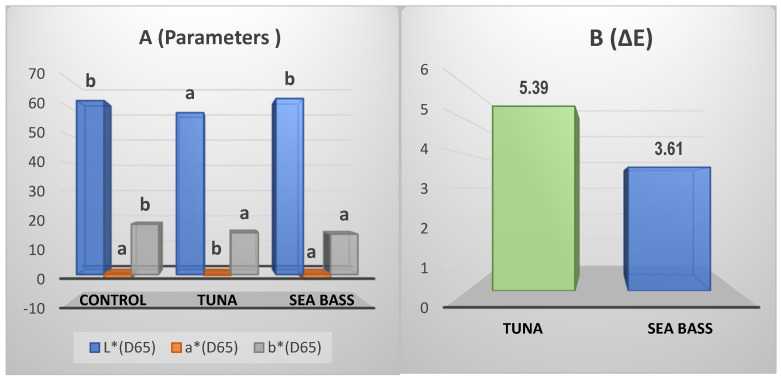
Color parameters for each assayed pasta (**A**) and variation with respect to control (**B**). Lowercase letters show significant differences among each type of pasta (*p* < 0.05).

**Figure 4 foods-10-01889-f004:**
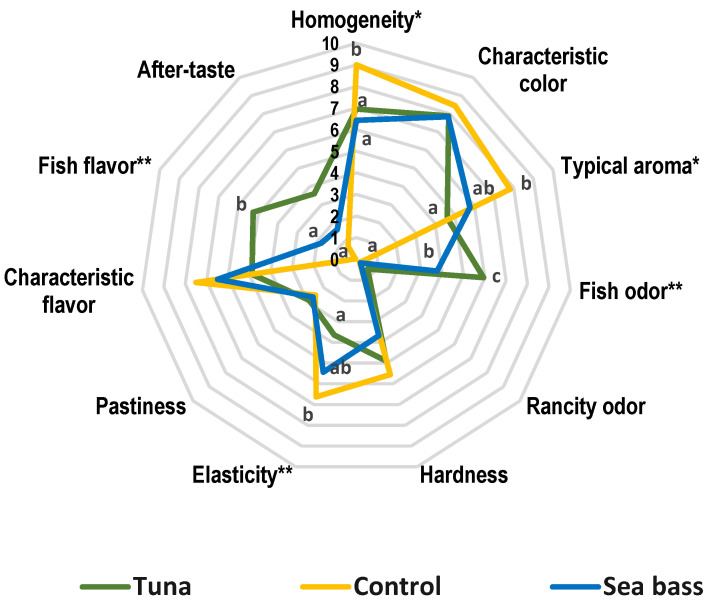
Radial graph of sensory attributes for enriched and control pasta. Lowercase letters show significant differences among the type of pasta. * Attribute showed significant differences (*p* < 0.05). ** Attribute showed high significant differences *(p <* 0.01).

**Figure 5 foods-10-01889-f005:**
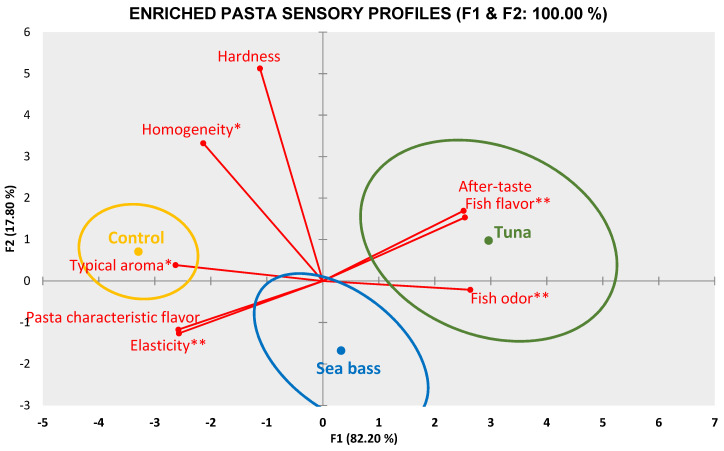
Sensory profile for fish enriched pasta and durum pasta (control). * Attribute showed significant difference (*p* < 0.05). ** Attribute showed significant difference (*p* < 0.01).

**Figure 6 foods-10-01889-f006:**
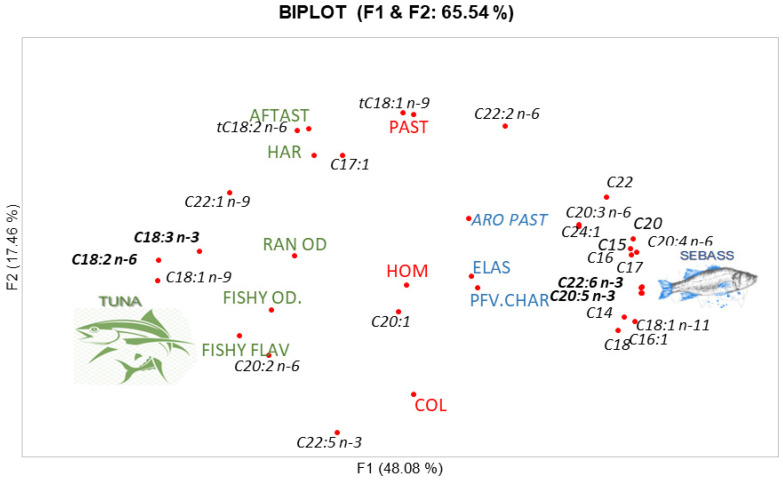
PCA from correlation matrix between sensory attributes and fatty acids composition for Sea bass pasta and Tuna pasta. AFTAST: after taste, HAR: hardness, RAN OD: rancidity odor, FISHY OD: Fishy odor, FISHY FLAV: Fishy flavor, PAST: Pastiness, ARO PAST: Pasta aroma, HOM: Homogeneity, COL: Typical color, ELAS: elasticity and PFV.CHAR. Pasta characteristic flavor.

**Table 1 foods-10-01889-t001:** Formulations used to make enriched pasta with fish by-products and their proximal analyses.

**Ingredients**	**Sea Bass Pasta (%)**	**Tuna Pasta (%)**	**Control Pasta (%)**
Durum wheat	72	72	75
Dried fish concentrate	3	3	0
Water	25	25	25
**Proximal Composition (%)**	**Sea Bass Pasta (%)**	**Tuna Pasta (%)**	**Control Pasta (%)**
Moisture	10.6	10.4	11.6
Protein	14.8	13.2	12.5
Fat	1.5	1.8	1.4
Fiber	1.23	1.27	1.20

**Table 2 foods-10-01889-t002:** Fatty acid profiles for enriched pasta with fish by-products.

Fatty Acids	Tuna Pasta	Sea Bass Pasta
C14	2.44 ± 0.09 b	0.96 ± 0.01 a
C15	0.53 ± 0.02 b	0.06 ± 0.09 a
C16	21.31 ± 0.10 b	18.04 ± 0.21a
C17	0.56 ± 0.01 b	0.13 ± 0.11a
C18	3.08 ± 0.09 b	2.65 ± 0.15 a
C20	0.38 ± 0.01 b	0.05 ± 0.09 a
C22	0.18 ± 0.00 b	0.04 ± 0.07 a
%SFA	28.48 b	21.93 a
C16:1	2.81 ± 0.40 b	1.49 ± 0.02 a
C17:1	0.31 ± 0.01	0.41 ± 0.36
tC18:1 n-9	0.05 ± 0.09	0.05 ± 0.09
C18:1 n-11	1.85 ± 0.03 b	1.60 ± 0.04 a
C18:1 n-9 (oleic)	12.82 ± 0.05 a	20.75 ± 0.57 b
C20:1	2.09 ± 0.28	2.01 ± 0.70
C22:1 n-9	0.00 ± 0.00	0.16 ± 0.04
C24:1	0.12 ± 0.10	0.00 ± 0.00
%MUFA	20.04 a	26.47 b
C18:3 Ω-3 (ALA)	0.57 ± 0.03 a	2.98 ± 0.68 b
tC18:2 n-6	0.00 ± 0.00	0.05 ± 0.04
C18:2 n-6 (linoleic)	38.00 ± 0.38 a	43.97 ± 0.09 b
C20:2 n-6	0.30 ± 0.18	0.48 ± 0.02
C20:3 n-6	0.07 ± 0.06	0.00 ± 0.00
C22:2 n-6	0.09 ± 0.08	0.04 ± 0.02
C20:4 n-6	0.59 ± 0.02 b	0.15 ± 0.12 a
C22:6 Ω-3 (DHA)	7.31 ± 0.23 b	2.05 ± 0.12 a
C20:5 Ω-3 (EPA)	3.64 ± 0.10 b	1.44 ± 0.05 a
C22:5 Ω-3	0.43 ± 0.01	0.44 ± 0.02
%PUFA	50.43	51.59
ΣΩ3	11.37 b	6.91 a
ΣΩ6	39.06 a	44.69 b
P/S ratio	1.77 a	2.35 b
Ω6/Ω3 ratio	3.43 a	6.47 b
mg Ω3/100 g	202.42 b	111.88 a
%DRI (EFSA)	80.97 b	44.75 a

SFA: Saturated Fatty Acids, MUFA: Monounsaturated fatty acids, PUFA: polyunsaturated fatty acids, P/S ratio: PUFA/SFA ratio, %DRI: dietary reference intake. Lowercase letters show significant differences between both types of pasta (*p* < 0.05).

**Table 3 foods-10-01889-t003:** TPA parameters for both types of enriched pasta and a control.

	HDN	ADH	SPG	COH	GUM	CHW	FRT
**Control pasta**	3726.35 ± 252.65 c	−16.02 ± 30.05 b	0.77 ± 0.09	0.68 ± 0.04 b	2545.55 ± 286.53 b	1993.26 ± 300.04 b	758.42 ± 86.63 b
**Sea bass pasta**	3206.25 ± 296.84 b	−13.79 ± 35.44 b	0.77 ± 0.09	0.49 ± 0.13 a	1610.34 ± 338.32 a	1279.20 ± 296.17 a	881.13 ± 125.81 c
**Tuna pasta**	2418.04 ± 304.11 a	−36.87 ± 20.34 a	0.72 ± 0.08	0.66 ± 0.06 b	1611.40 ± 291.77 a	1156.31 ± 239.39 a	479.18 ± 93.53 a

HDN: Hardness, ADH: Adhesiveness, SPG: springiness, COH: cohesiveness, GUM: Gumminess, CHW: chewiness, FRT: fracturability. Lowercase letters show significant differences among each type of pasta (*p* < 0.05).

**Table 4 foods-10-01889-t004:** Values of technological properties for each enriched pasta developed.

	WG (%)	SI (g/g)	CL (%)	M (%)
**Control**	167.06 ± 3.15 b	3.20 ± 0.03 c	4.46 ± 0.17 a	11.59 ± 0.38 b
**Sea bass**	128.70 ± 8.80 a	2.72 ± 0.15 b	5.14 ± 0.75 b	10.58 ± 0.64 a
**Tuna**	129.35 ± 11.25 a	1.57 ± 0.04 a	4.91 ± 0.19 b	10.42 ± 0.21 a

WG: weight gain, SI: swelling index, CL: cooking losses, M: moisture. Lowercase letters show significant differences among types of pasta (*p* < 0.05)

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
