# Peer review of "Innovative Development of Pasta with the Addition of Fish By-Products from Two Species"

_foods, 2021, doi:10.3390/foods10081889_

Round 1

Reviewer 1 Report

I have few common concerns related to the manuscript.

Authors neglect how spread out the data is. The manuscript is weak in terms of statistical approach. Standard deviation (SD) should be used to measure variability used in statistics.

Table 1. The proximate composition is not complete. Soluble and insoluble fibers make up the two basic categories of dietary fiber. Fibers affect the consistency, texture, rheological behavior, and sensory attributes of the end products. 

Please give more details about all the materials and methods presented in this manuscript. Few details make it difficult to reproduce the methods.

The results and discussion part only focus on analyzing phenomena and lack profound explanations, making the overall article lack of depth and deep meaning.

Author Response

Question

Response

Page

1.1 The manuscript is weak in terms of statistical approach. Standard deviation (SD) should be used to measure variability used in statistics.

Thank you very much for your reccomendation. All the data shown in the results are statistically analyzed using an ANOVA, which shows the significant differences between means. However, the stardard desviation has been added in all tables of the results.

p.5/6/7/10

1.2 Table 1. The proximate composition is not complete. Soluble and insoluble fibers make up the two basic categories of dietary fiber.

Thank you very much for your advice. The fiber content is very low in this type of pasta. Durum wheat pastas has around 2% fiber and, on the other hand, both sea bass and tuna hardly provide fiber, so the enriched pasta maintains that low percentage of fiber.

p.3

1.3 Please give more details about all the materials and methods presented in this manuscript.

Thank you very much for your observation. Certain data about the samples to carry out the methods involved have been incorporated. In addition, the pertinent information has been added in those analyzes that were not expressly described.

p.2-5

1.4 The results and discussion part only focus on analyzing phenomena and lack profound explanations.

Thank you very much for your advice. We, the authors, have tried to discuss the results with other references in the bibliography and in our opinion we have done it.

p.5-15

Reviewer 2 Report

Paragraph 2.2 (lines 74-81) – please specify dough moisture or give the moisture content of all raw materials used for pasta production. What was the form (shape) of the pasta? There is also a lack of information about the control sample (which is used for comparative purposes later in the manuscript)

Table 1. Add proximal composition for the control sample.

Line 91 - give reference for this method.

Line 107-115 – whether the color of uncooked or cooked pasta was assessed?

Formula 1 - Is "standard pasta" the same as "control pasta"? If yes, please change in formula description "standard" to "control pasta". If not, how was standard pasta chosen, please specify?

Line 119 – Is “ideal cooking time” the same as “optimum cooking time” – please use the same name throughout the manuscript.

Line 119-120 – How the pasta was dried? Do you mean drying the surface of the pasta or completely drying it?

Formula 2– The description can be misleading. Whether "Weight of pasta after drying" refers to cooked pasta after drying (see line 119) or uncooked pasta (after production see line 80)?

Formula 3 – also in this case description can be misleading.

Line 196 – change ω to Ω

Figure 1 – Why hardness is expressed in "kg" not in "kg/mm*s2" as was shown in the Materials and methods section (see line 95)

Line 216 – “in”  ?

Table 3 and 4 - I see problems with the layout- the top rows (header cells) are invisible. (please check)

Table 4 – why the moisture content of enriched pasta given in this table is different from the values shown in table 1

Figure 6 - A caption should be below the illustration

Line 341 – “As expected, pasta with fish concentrates had a higher content of fatty acids, although it had some exceptions” – Higher content compare to …? Control pasta? to confirm this, the fatty acid content in the control sample should be given. What exceptions?

Line 424- “contribution of protein” to confirm this the content of protein in fish by-product should be given. In table 1 only protein content in enriched pasta is given so it is difficult to determine to what extent the introduction of fish products changed the protein content in pasta since its content was not given in the control sample.

Author Response

Question

Response

Page

2.1 please specify dough moisture or give the moisture content of all raw materials used for pasta production. What was the form (shape) of the pasta? There is also a lack of information about the control sample

Thank you very much for you reccomendation. The moisture content of all raw materials was incorporated in the section 2.2. The form used was fusilli format as we have indicated in the same section. On top of that, the necessary information about control sample have been included in Table 1.

p.3

2.2 Table 1. Add proximal composition for the control sample. 

Thank you for your reccomendation. Proxumal composition for the control sample was added in the Table 1.

p.3

2.3 Line 91 - give reference for this method.

Thank you very much for your advice. The method used  was found in the own instruction manual of the texturometer.

p.3

2.4 Line 107-115 – whether the color of uncooked or cooked pasta was assessed?

Thank you for your question. The analyse was evaluated in cooked pasta according to its optimal cooking time.

p.3

2.5 Formula 1 - Is "standard pasta" the same as "control pasta"? If yes, please change in formula description "standard" to "control pasta". If not, how was standard pasta chosen, please specify?

Thank you for your observation. Stardard pasta and control pasta is the same. This description has been changed.

p.3

2.6 Line 119 – Is “ideal cooking time” the same as “optimum cooking time” – please use the same name throughout the manuscript.

Thank you very much for your advice. These terms are the same and have been modified in the manuscript.

p.3/4

2.7 Line 119-120 – How the pasta was dried? Do you mean drying the surface of the pasta or completely drying it?

Thank you very much for your question. Pasta was dried on the surface with absorbent paper. This question has been clarified in this section.

p.4

2.8 Formula 2– The description can be misleading. Whether "Weight of pasta after drying" refers to cooked pasta after drying (see line 119) or uncooked pasta (after production see line 80)?

Thank you for your observation. "Weight of pasta after drying" refers to cooked pasta after drying. We have modified the description to clarify it by adding the word "cooked".

p.4

2.9 Formula 3 – also in this case description can be misleading.

Thank you very much. We have changed this description in the same way.

p.4

2.10 Line 196 – change ω to Ω

Thank you very much for your observation. This symbol has been modified.

p.6

2.11 Figure 1 – Why hardness is expressed in "kg" not in "kg/mm*s2" as was shown in the Materials and methods section (see line 95)

Thank you for your question. It was a mistake, hardness is expressed in kg/mm*s2, we have already changed it to the correct unit.

p.7

2.12 Table 3 and 4 - I see problems with the layout- the top rows (header cells) are invisible.

Thank you very much for your observation. Both tables have been corrected.

p.7/10

2.13 Table 4 – why the moisture content of enriched pasta given in this table is different from the values shown in table 1

Thank you for your question. The correct values are in Table 4. It was a failure in the decimals that has already been modified in Table 1.

p.10

2.14 Figure 6 - A caption should be below the illustration

Thank you for your observation. It had moved over the image. We have put it in its correct place, below the figure.

p.12

2.15 Line 341 – “As expected, pasta with fish concentrates had a higher content of fatty acids, although it had some exceptions” – Higher content compare to …? Control pasta? to confirm this, the fatty acid content in the control sample should be given. What exceptions?

Thank you very much for your questions. The purpose of this study is the comparative between two pasta with two species in terms of omega 3 fatty acids. Therefore, it is a mistake that has been clarified and that aims to demonstrate the greater amount of this type of fatty acids in pasta with tuna compared to pasta with sea bass. It is considered that control has an insignificant content of these fatty acids and so, it is not comparable. We only compared control in terms of technological and sensory properties, since in this case it could be compared.

p.12

2.16 Line 424- “contribution of protein” to confirm this the content of protein in fish by-product should be given. In table 1 only protein content in enriched pasta is given so it is difficult to determine to what extent the introduction of fish products changed the protein content in pasta since its content was not given in the control sample.

Thank you for your observation. The content of protein of control pasta to compare with the enriched pasta and show this contribution related to fish concentrates have been added in Table 1.

p.3/14

Round 2

Reviewer 1 Report

All the suggestion have been addressed.